# Impaired Pharmacokinetics of Amiodarone under Veno-Venous Extracorporeal Membrane Oxygenation: From Bench to Bedside

**DOI:** 10.3390/pharmaceutics14050974

**Published:** 2022-04-30

**Authors:** Mickaël Lescroart, Claire Pressiat, Benjamin Péquignot, N’Guyen Tran, Jean-Louis Hébert, Nassib Alsagheer, Nicolas Gambier, Bijan Ghaleh, Julien Scala-Bertola, Bruno Levy

**Affiliations:** 1Service de Médecine Intensive et Réanimation, Centre Hospitalier Régional Universitaire de Nancy (CHRU Nancy), Hôpital Brabois, 54000 Nancy, France; bpequignot@hotmail.fr (B.P.); blevy5463@gmail.com (B.L.); 2Groupe Choc, Équipe 2, INSERM U 1116, Faculté de Médecine, 54000 Nancy, France; 3Faculté de Médecine, Université de Lorraine, 54000 Nancy, France; nguyen.tran@univ-lorraine.fr; 4Laboratoire de Pharmacologie, Assistance Publique des Hôpitaux de Paris (AP-HP), Hôpitaux Universitaires Henri Mondor, Université Paris Est-Créteil, 94000 Créteil, France; claire.pressiat@aphp.fr; 5Team 3, INSERM U955, Université Paris Est Créteil, Université Paris-Est, 94010 Créteil, France; 6UMR S955, DHU A-TVB, Université Paris-Est Créteil (UPEC), Université Paris-Est, 94000 Créteil, France; 7École de Chirurgie, Faculté de Médecine, Université de Lorraine, 54000 Nancy, France; 8Institut de Cardiologie, Hôpital Pitié-Salpêtrière, CHU Pitié-Salpêtrière, AP-HP, Université de la Sorbonne, Boulevard de L’Hôpital, 75013 Paris, France; jean.l.hebert@gmail.com; 9Centre Hospitalier Régional Universitaire de Nancy (CHRU Nancy), Service de Pharmacologie Clinique et Toxicologie, Université de Lorraine, 54000 Nancy, France; nassibalsagheer@gmail.com (N.A.); nicolas.gambier@univ-lorraine.fr (N.G.); j.scala-bertola@chru-nancy.fr (J.S.-B.); 10CNRS, IMoPA, Université de Lorraine, 54000 Nancy, France; 11U955-IMRB, Inserm, Université Paris-Est Créteil (UPEC), École Nationale Vétérinaire d’Alfort, Maisons-Alfort, 94000 Créteil, France; bijan.ghaleh@inserm.fr

**Keywords:** amiodarone, VV ECMO, pharmacokinetics, Monte-Carlo simulations

## Abstract

Background: Adjusting drug therapy under veno-venous extracorporeal membrane oxygenation (VV ECMO) is challenging. Although impaired pharmacokinetics (PK) under VV ECMO have been reported for sedative drugs and antibiotics, data about amiodarone are lacking. We evaluated the pharmacokinetics of amiodarone under VV ECMO both in vitro and in vivo. Methods: In vitro: Amiodarone concentration decays were compared between closed-loop ECMO and control stirring containers over a 24 h period. In vivo: Potassium-induced cardiac arrest in 10 pigs with ARDS, assigned to either control or VV ECMO groups, was treated with 300 mg amiodarone injection under continuous cardiopulmonary resuscitation. Pharmacokinetic parameters C_max_, T_max_ AUC and F were determined from both direct amiodarone plasma concentrations observation and non-linear mixed effects modeling estimation. Results: An in vitro study revealed a rapid and significant decrease in amiodarone concentrations in the closed-loop ECMO circuitry whereas it remained stable in control experiment. In vivo study revealed a 32% decrease in the AUC and a significant 42% drop of C_max_ in the VV ECMO group as compared to controls. No difference in T_max_ was observed. VV ECMO significantly modified both central distribution volume and amiodarone clearance. Monte Carlo simulations predicted that a 600 mg bolus of amiodarone under VV ECMO would achieve the amiodarone bioavailability observed in the control group. Conclusions: This is the first study to report decreased amiodarone bioavailability under VV ECMO. Higher doses of amiodarone should be considered for effective amiodarone exposure under VV ECMO.

## 1. Introduction

The use of veno-venous extracorporeal membrane oxygenation (VV ECMO) has increased since the 2009 H1N1 pandemic, and might improve survival up to 70% in COVID-19 acute respiratory distress syndrome (ARDS) [1]. Nevertheless, VV ECMO was made responsible for impairing drug pharmacokinetics (PK), due to a possible modification of distribution volumes as well as to potential physicochemical interactions between drugs and ECMO circuitry, especially in case of highly lipophilic molecules [2]. Since the 1970s, membrane oxygenators (MO) have been progressively modified to make them smaller and more efficient, and their compounds now consist of molecules such as polymethylpentane (PMP) instead of silicone. PMP is a microporous polymer with a thin non-porous matrix on the blood side that requires diffusion and pressure gradients for molecules to pass through, which would minimize the amount of plasma leakage through the membrane. Harthan et al. reported that, despite the use of newer components in ECMO circuits, a large amount of medication is adsorbed into the circuit [3]. Absorption is driven by the electrostatic and hydrophobic interactions. Electrostatic interactions dominate when surface coatings are applied to the circuit, whereas the hydrophobic interactions tend to dominate when lipophilic drugs adhere to the tubes and membrane oxygenator without coating [4]. Wildschut et al. demonstrated that drug absorption is positively correlated with the degree of lipophilicity and that octanol/water partition coefficient (LogP) values could predict increased drug loss [5]. The effect of ECMO on drugs PK has been studied for sedative and anti-infective medications and Raffaeli et al. reported in their in vitro study a substantial drug loss in the extracorporeal membrane oxygenation circuits within 24 h of use, for the following molecules: paracetamol 49%, morphine 51%, midazolam 40%, fentanyl 84%, sufentanil 83% [6]. Since amiodarone is one of the most lipophilic drugs (LogP = 7.58), the issue of a possible influence of ECMO circuits on its bioavailability may rise.

To the best of our knowledge, only pediatric sparse case reports have shown therapeutic failure of its delivery at conventional posology during shockable cardiac arrest, whereas amiodarone PK under VV ECMO has never been studied in vivo so far [7,8]. It is a matter of the utmost importance because amiodarone is one of the very few molecules that can be safely used in intensive care units for treating both cardiac arrhythmias and shockable cardiac arrest [9]. The incidence of paroxysmal atrial fibrillation (AF) is high in critically ill patients [10]. This atrial arrhythmia has been associated with worsen outcome in patients with ARDS [11]. The loss of atrial function induces a 20% drop in cardiac output, which impairs oxygen delivery [12]. Recently, Li et al. reported that new-onset atrial arrhythmias are a frequent complication during VV ECMO and are independently associated with odds ratio for in-hospital mortality as high as 2.21 CI_95_ [1.08–4.55], with an interesting early temporal association of atrial arrhythmias with ECMO initiation (median time to onset of 1.7 days after ECMO deployment) [13,14]. Otherwise, the worst conditions clinicians would have to manage are refractory ventricular arrhythmias and cardiac arrest. According to the 2018 American Heart Association (AHA) guidelines for cardiopulmonary resuscitation (CPR), intravenous (IV) amiodarone 300 mg bolus should be administered during shockable cardiac arrest refractory to 3 consecutive electrical cardioversions, occurring apart from VV ECMO conditions [15]. In addition, Mc Daniel et al. showed in their ex vivo model, a rapid and heavy extraction of amiodarone by ECMO circuitry during the first 30 min of the procedure [16]. We postulated that CPR under VV ECMO would represent the worst impairment of amiodarone PK although this situation requires early high concentrations at the aortic root to ensure efficient receptor-ligand interactions and to improve the pharmacodynamics (PD) of amiodarone. In line with these observations, the present study aimed to assess the role of VV ECMO on amiodarone PK using both an in vitro model and an in vivo ARDS porcine CPR model.

## 2. Materials and Methods

### 2.1. In Vitro Study

A closed ECMO circuit has been assembled, which consisted of a console (Rotaflow Console, Maquet, Rastatt, Germany), a centrifugal pump (Rotaflow Centrifugal Pump System, Maquet, Germany) and a circuit tubing together with a membrane oxygenator (PLS-I oxygenator, Maquet, Germany), which was itself bound to a mechanical gas blender system (Sechrist Model 20090, Sechrist, Anaheim, CA, USA). Our protocol was based on the experimental data published by Raffaeli et al. [6]. Closed circuits were assembled by connecting arterial and venous lines together, allowing a continuous flow of priming fluid throughout the entire circuit. A specific buffer solution composed of 770 mL Dulbecco’s Phosphate Buffered Saline (PBS) (Life Technologies Corporation, Grand Island, NE, USA) at pH 7.4 was used for priming the ECMO circuit, which was filled to maximal capacity. Flow rates were set to run for 6 h at 3.5 L/min (i.e., pump rotation speed at 3500 rpm). Temperature of the fluid was maintained at 37 °C via a specific heater-cooler unit connected to the ECMO circuitry.

Drugs were injected through a line which was connected to the circuitry. The line was flushed with 5 mL of physiologic saline solution (0.9%). The experiment was separately performed with both 100 mg and 300 mg amiodarone hydrochloride injection formulation (150 mg/3 mL) (Sanofi-Aventis, Paris, France) diluted in 10 mL of glucose 5% solution administered via the pre pump admission line. Once the circuit was primed, and before drug injection, a pre-membrane sample was obtained from a line attached to the tubing. During the circuit run, pre-membrane 1 mL samples were drawn at the following time intervals throughout the 24 h in vitro testing period: 2, 10, 30 min, then 1, 2, 3, 4, 5, 6 and 24 h after amiodarone injection in the ECMO circuitry. The experiment was performed thrice for amiodarone 100 mg and control groups, and once for the amiodarone 300 mg group.

Spontaneous drug degradation at 2, 10, 30 min, then at 1, 2, 3, 4, 5, 6, and 24 h was assessed with a specific control experiment performed as follows: PBS 770 mL was warmed at 37 °C in a specific temperature-controlled magnetic stirring container. Amiodarone 100 mg was injected, and subsequent samples were drawn. Data were presented as mean (±SD) or percentage. All experiments were conducted in triplicate.

### 2.2. In Vivo Study

#### 2.2.1. Animal Model

All experiments were reviewed and approved by the Nancy University Ethics Committee for Animal Experimentation (APAFIS number 26921, available on 21 September 2021). The procedure for the care and sacrifice of study animals was in accordance with the European Community Standards on the Care and Use of Laboratory Animals.

#### 2.2.2. Animal Preparation

Experiments were carried-out on domestic male adult pigs (Landrace) weighing 45 to 65 kg. Animals were fasted overnight with free access to water. Intramuscular premedication was performed with ketamine (1.5 mg/kg, Warner Lambert, Nordic, AB Solna, Sweden) before transportation to the experiment facility. Sedation was deepened with propofol 2.5 mg/kg (B Braun, Melsungen, Germany) via an ear vein cannula. After being placed in supine position, animals were intubated with a 7.5 mm internal diameter endotracheal tube (ETT). Immediately thereafter, they received an initial IV bolus of both propofol (2 mg/kg) (B Braun, Melsungen, Germany) and cisatracurium (0.5 mg/kg) (GlaxoSmithKline, Marly le-Roi, France), both followed by a continuous venous infusion (see below). An initial rapid IV infusion of 1000 mL normal saline solution was given after anesthesia induction. Soon after orotracheal intubation, pigs were ventilated under assisted volume-controlled mode with a tidal volume of 6 mL/kg and FiO_2_ 1.0. The ventilator settings were then adjusted according to the results of blood gas analyses performed both at basal state and after shock induction.

A fluid-filled pressure catheter (Seldicath, Plastimed Prodimed, Neuilly en Thelle, France) was surgically inserted into the left carotid artery for systemic blood pressure monitoring. The right carotid artery was dissected, and a Transit Time Flow probe (Transonic Systems Inc., Ithaca, NY, USA) was secured around it. Data were computed using a designated analysis program (IOX 2.4.2.6^®^, EMKA Technologies, Paris, France). A central venous line was surgically inserted into the left internal jugular vein for maintaining anesthesia with a continuous infusion of the following drugs: propofol (6 to 8 mg/kg/h), fentanyl (0.01 mg/kg/h) (Pharmalink, Stockholm, Sweden), and cisatracurium (0.5 mg/kg/h). A specific catheter was inserted into the thoracic ascending aorta via a 6Fr introducer sheath previously advanced under ultrasound guidance via the femoral artery and was dedicated for central blood samplings.

#### 2.2.3. ARDS Induction

According to Araos et al., induction of a “double hit” lung injury was performed by repeated lung lavages (30 mL/kg warm 0.9% saline solution intratracheally at 38.5 °C) until PaO_2_/FiO_2_ fell below 250 mmHg, followed by two hours of injurious ventilation with PEEP 0 cm H_2_O and inspiratory pressure up to 40 cm H_2_O, RR 10 bpm, inspiratory to expiratory time ratio (I:E), 1:1, together with permanent 100% FiO_2_ [17].

#### 2.2.4. Experiment Protocol

VV ECMO implantation: once required, VV ECMO canulation was performed. The venous outflow cannula (15Fr, Biomedicus cannulae, Medtronic, Minneapolis, MN, USA) was surgically inserted into the right internal jugular vein. A venous inflow cannula (21Fr, Biomedicus cannulae, Medtronic), was percutaneously inserted into the femoral vein after ultrasound-guided venous puncture and guiding-wire insertion. A 100 UI/kg dose of Heparin (Héparine Sodique Choay, Sanofi-Aventis, Paris, France) was administered prior to the cannulation phase. The ECMO circuitry consisted of a console (Rotaflow Console, Maquet, Germany), a centrifugal pump (Rotaflow Centrifugal Pump System, Maquet, Germany) and a circuit tubing together with a membrane oxygenator (PLS-I oxygenator, Maquet, Germany), linked to a mechanical gas blender system (Sechrist Model 20090, Sechrist, Anaheim, CA, USA). The oxygen/air blender and the sweep gas flow were adjusted to maintain PO_2_ and PCO_2_ in the ranges of 75 mmHg to 100 mmHg and of 30 mmHg to 48 mmHg, respectively, before initiation of the experimentation. The VV ECMO circuit was primed with 770 mL of 0.9% saline solution and pump rotation speed was adjusted to maintain ECMO blood flow between 3.5 and 4.0 L/min.

After animal preparation and ARDS induction, pigs were allocated whether to the control or to the VV ECMO group. Cardiac arrest was achieved in the whole population by iterative potassium chloride injections. CPR was immediately started by using an automatic chest compression device (LUCAS II, Lund University Cardiopulmonary Assist System; Physio-Control Inc./Jolife AB, Lund, Sweden) which standardized cardiac massage at 110 compressions per minute, with 50% duty cycle and compression depth adjusted as a function of the sternum height. The quality of CPR was assessed by monitoring the central arterial pressure. An amiodarone 300 mg bolus diluted in 10 mL of 5% glucose solution was administered after cardiac arrest induction. According to previously published data about monitoring amiodarone PK during CPR in swine models [18,19,20], serial blood specimens (10 mL) were collected from the aortic arterial line, at 30, 60, 90, 120, 150, 180, 240, 300 s, and at 6, 8, 10, and 12 min after amiodarone injection. Prior to each specimen collection, 10 mL of blood were aspirated and discarded to avoid any dilution effect or contamination. After each specimen had been collected, 10 mL of 0.9% normal saline solution were injected to maintain arterial line patency. Blood specimens were placed in lithium heparin collection tubes, which were immediately centrifuged for 15 min at 1800× *g*. Separated plasma was pipetted into duplicate 2 mL microcentrifuge tubes and frozen to a temperature of −80 °C in a laboratory grade freezer. The in vivo experiment is schematically illustrated in Figure 1.

### 2.3. Samples Analysis

Amiodarone was assayed with Sciex QTrap^®^ 4500 liquid chromatography tandem-mass spectrometry (LC-MS/MS) (Sciex, Villebon-sur-Yvette, France) using the Chromsystems commercial method for antiarrhythmic drugs (Chromsystems, Gräfelfing, Germany). Briefly, 50 μL of samples were added with 25 μL of extraction buffer and 250 μL of deuterated internal standard. After vortexing and centrifugation of the mixture, the obtained supernatant was 10-fold diluted and 10 μL of the sample was injected in the LC-MS/MS system. The quantification limit was 70 μg/L. Intraday precision ranged from 3.9% to 4.5%, whereas inter-day precision ranged from 5.3% to 7.4%. Accuracy ranged from −9.3% to 9.9%. No matrix interference was noticed for amiodarone with both porcine plasma and Dulbecco’s Phosphate Buffered Saline.

### 2.4. Pharmacokinetic Analysis

#### 2.4.1. Observed Amiodarone Pharmacokinetic Parameters

The following pharmacokinetic parameters maximum concentration (C_max_), time to maximum concentration (T_max_) and area under the curve (AUC) were directly obtained by observation of amiodarone concentration profiles. The values of AUC were calculated using the trapezoidal rule over the in vivo experiment period. Bioavailability in VV ECMO group was calculated as follows:$$F\left(\%\right)=\frac{{\mathrm{AUC}}_{\mathrm{VV}\;\mathrm{ECMO}}}{{\mathrm{AUC}}_{\mathrm{CONTROL}}}\times \frac{{\mathrm{Dose}}_{\mathrm{CONTROL}}}{{\mathrm{Dose}}_{\mathrm{VV}\;\mathrm{ECMO}}}\times 100$$

#### 2.4.2. Estimated Amiodarone Pharmacokinetic Parameters by PK Modeling

The model was developed using a non-linear mixed-effect modeling approach (Monolix^®^ version 2020.R1 (Available online: www.lixoft.eu (accessed on 2 February 2021)). For PK modeling, only concentrations from peak to valley were considered. Parameters were estimated using the stochastic approximation expectation maximization (SAEM) algorithm. One, two and three-compartment structural models with first-order elimination were tested for defining the basic structural model. Categorial covariates were tested as follows:*θ_i_* = *θ_pop_* × *θ^Cov^*(1)
where *θ_i_* is the individual parameter (elimination clearance: CL, volume of distribution of the central compartment: Vc, inter-compartment clearance: Q, and apparent volume of distribution of the peripheral compartment: Vp) for the *i*th patient, *θ_pop_* is the typical value of the parameter, *Cov* is the category 0 or 1 for the covariate under study, and *θ^Cov^* is the covariate parameter.

Continuous covariates were associated with PK parameters by a power function as follows:(2)$$\theta_{i}=\theta_{pop}\times {\left(\frac{Cov_{i}}{\left(Median\left(Cov\right)\right)}\right)}^{PWR}$$
where *Cov_i_* is the covariate value for the *i*th patient and the *PWR* exponent is the power parameter.

An effect of a covariate on a structural parameter was retained if it caused a decrease in the Bayesian information criterion (BIC) and/or reduced the corresponding between subject variability (BSV) with *p* < 0.05. The objective function value reduction was tested for significance via a likelihood ratio test. Diagnostic graphics were used for evaluating the goodness-of-fit. Concentration profiles were simulated and compared to the observed data with the aid of the predicted-corrected visual predictive check in order to validate the model (PC-VPC). Empirical percentiles (percentiles of the observed data [5th, 50th and 95th], calculated either for each unique value of time, or pooled by adjacent time intervals) and theoretical percentiles (percentiles of simulated data) were assessed graphically. Individual PK parameters (median, Q1–Q3), i.e., maximum concentration (C_max_), Area Under the Curve (AUC) and time to maximum concentration (T_max_), were determined from the model for both VV ECMO and control groups.

### 2.5. Statistical Analysis

In vitro study: One-way repeated ANOVA was used for comparing amiodarone concentrations in 100 mg amiodarone ECMO group and 100 mg amiodarone control group all along the experiment. Significance was considered for *p* value < 0.05.

In vivo study, individual PK parameters: Due to the experimental and pilot nature of the study, it was not possible to perform a sample size calculation. Mann–Whitney test was conducted to test significant differences between VV ECMO group and control group relative to hemodynamic measurements during CPR, maximum Concentration (C_max_), Area Under the Curve (AUC) and Time to maximum concentration (T_max_). Statistical significance was indicated by a *p*-value < 0.05. Statistical analysis was performed using R version 4.0.1 for MacOS^®^ (https://www.r-project.org/, accessed on 10 March 2020).

Monte-Carlo simulations and probability of target attainment. The PK model was used for performing a Monte-Carlo simulation of 1000 individuals achieving steady-state AUC values. Using our final model, we simulated different dosing schemes for amiodarone (1000 Monte-Carlo simulations) in animals on VV ECMO. The results were compared graphically by representing the amiodarone AUC obtained in the control group.

## 3. Results

### 3.1. In Vitro Experimentation

In the ECMO groups, amiodarone concentration decreased from 99.0% to 99.6% of its initial maximal concentration (53.8 mg/L and 201.0 mg/L) after 120 min in 100 mg and 300 mg amiodarone ECMO groups, respectively. Amiodarone concentrations obtained through the first 2 h period of the in vitro experiment are presented in Figure 2. A potential later release of amiodarone in the ECMO groups after the first 2 h was not observed and amiodarone concentration continued to decrease until 24 h. In the 100 mg amiodarone control group, amiodarone concentration slightly increased from 42.3 mg/L to 50.7 mg/L between 0 and 30 min and then reached a plateau with stable concentrations during the following 23.5 h. One-way repeated ANOVA revealed a significant difference in amiodarone concentrations between the 100 mg amiodarone ECMO group and the 100 mg amiodarone control group from 10 min (*p* = 0.03) to 24 h (*p* < 0.01).

### 3.2. In Vivo Experimentation

#### 3.2.1. Population

Experiments were performed in 10 pigs. After animal preparation, 5 pigs received VV ECMO, and 4 pigs were allocated to the control group. One pig of the control group died before time of measurements. The Mann–Whitney test found no significant difference between ECMO group and control group relative to basal weight, heart rate (HR), systolic arterial pressure (SAP), diastolic arterial pressure (DAP), mean arterial pressure (MAP), indicating the groups were equivalent for these variables (Table 1). Mann–Whitney test found no significant difference between the groups relative to mean arterial pressure or mean carotid blood flow during CPR, indicating comparable quality of mechanical chest compression between groups.

#### 3.2.2. Observed Amiodarone Pharmacokinetic Parameters

A total of 108 serum samples from nine pigs were collected and analyzed over the 12 min CPR period. Amiodarone mean concentration profiles observed in ECMO and control groups are plotted in Figure 3. Observed mean time to maximum concentration (T_max_) and maximum concentration (C_max_) are presented in Table 2. Area under the curve was calculated using trapezoidal rule over the 12 min period of the in vivo experiment (AUC_0__→__12min_). If no difference was observed for T_max_ between the ECMO and the control groups, a lower C_max_ and a statistically lower AUC_0__→__12min_ was observed in the ECMO group in comparison to the control group. From the calculated AUC_0__→__12min_, the ECMO group exhibited a bioavailability of 67.1% for amiodarone in comparison to control group.

#### 3.2.3. Estimated Amiodarone Pharmacokinetic Parameters

A two-compartment model with first-order distribution and elimination was able to accurately describe amiodarone plasma concentrations. The pharmacokinetic parameters of this model were clearance (CL), central volume of distribution (Vc), intercompartmental clearance (Q), peripheral volume of distribution (Vp). Residual variability was described using a proportional error model for amiodarone plasma concentrations. Interindividual variability was retained for CL and Vc. Amiodarone PK parameters were influenced by the ECMO covariable. Table 3 summarizes the final population pharmacokinetic estimates for the model, including the relative standard errors (RSE). All parameters were well estimated given the low RSE values (<35%). Goodness of fit plots presented in Appendix A comfort the accuracy of the model. PC-VPC of the final model showed that the 5th, 50th and 95th percentiles of observed data were clearly included within the 90% confidence interval (CI) of the 5th, 50th and 95th simulated percentiles for plasma amiodarone concentrations (Figure 4).

The final covariate model was:Cl = 0.39 × (0.69)^ECMO^
Vc = 1.09 × (1.1)^ECMO^

Clearance was 0.39 L/h and 0.27 L/h in the non-ECMO and ECMO group, respectively, resulting in a 31% lower clearance in the ECMO group.

Similarly, central distribution volume was 1.09 L and 1.2 L in the non-ECMO and ECMO group, respectively, resulting in a 10% increased distribution volume in the ECMO group.

Individual pharmacokinetic parameters estimated from the model are presented in Table 4. The Mann–Whitney test revealed a non-significant but relevant reduction in the AUC over 12 min between VV ECMO and control groups. The AUC in the control group was larger than in the VV ECMO group with median value (Q1–Q3) of 698 min·mg/L (630–892) vs. 400 min·mg/L (365–491), respectively (*p* = 0.06). The Mann–Whitney U test indicated significant differences relative to C_max_ between VV ECMO and control groups. The C_max_ in the control group was significantly higher than in the VV ECMO group with median value (Q1–Q3) of 123.5 mg/L (109.5–150.0) versus 61.7 mg/L (55.3–80.3), respectively (*p* = 0.02). The median (Q1–Q3) C_max_ for each group is summarized in Table 2. The median (Q1–Q3) T_max_ for each group is summarized in Table 2. The Mann–Whitney U test indicated there was no significant difference for the T_max_ between the control group and the VV ECMO group, at 90 s (75–90) and 90 s (60–90), respectively (*p* = 1). From the estimated AUC, the ECMO group exhibited a mean relative bioavailability of 57.3% for amiodarone in comparison to control group.

#### 3.2.4. Monte-Carlo Simulations of Pharmacokinetics

As the median AUC in the control group was 698 min·mg/L (630–892), several simulations for as sessing the amiodarone dose required to target the same AUC under VV ECMO were performed. According to the present model, amiodarone 600 mg should be efficient under VV ECMO to achieve the AUC of the control group (Figure 5).

## 4. Discussion

To the best of our knowledge, this is the first study reporting pharmacokinetics of amiodarone under VV ECMO in both in vitro and in vivo experiments. In presence of VV ECMO, in vitro study led in the present work found a decrease in amiodarone concentration after the first two minutes and a significant and rapid decrease in the same concentrations after the tenth minute. The use of 100 mg or 300 mg dose of amiodarone led to the same observations. Because no spontaneous degradation of amiodarone was observed under similar control experimental conditions, it can be unequivocally stated that the drop in amiodarone concentration was the consequence of VV ECMO. As already described for other lipophilic drugs [3], we can hypothesize that the decrease in amiodarone concentration in our study can probably be related to an adsorption phenomenon on the VV ECMO materials. Indeed, the main determinants usually described for drug adsorption during ECMO are molecular weight, ionized fraction and lipophilicity [2]. As amiodarone is one of the most lipophilic drugs (LogP = 7.58) and has a high protein binding (96%), the hypothesis of its adsorption on VV ECMO materials can be rationally considered. Nevertheless, despite the frequent use of amiodarone in intensive care for the control of cardiac arrhythmias or shockable cardiac arrest, we did not find any clinical or preclinical study in the literature addressing the modification of amiodarone pharmacokinetics under VV ECMO. To the best of our knowledge, only McDaniel et al. published an in vitro study in 2021 reporting amiodarone extraction through the ECMO circuit [8]. These authors found significant adsorption of amiodarone to both the membrane oxygenator and the circuit within the first few minutes of their experiment, as it was observed in our in vitro study. A sharp decrease in drug exposure was observed in all ECMO circuits tested, whether primed with blood or crystalloid. Compared with blood, the effect of ECMO on amiodarone exposure was higher in crystalloid-primed ECMO. Our in vitro study confirms the observations of McDaniel et al. and complements this previous study. Indeed, unlike McDaniel et al. we used only PBS solution and did not add albumin to the in vitro medium. Thus, we observed a decrease in amiodarone concentrations as rapid as that observed in the McDaniel et al. study performed with crystalloid-initiated ECMO, but of greater intensity (99% versus 80% after 1 h). This difference may probably be explained by the absence of albumin in our in vitro medium, thus limiting the nonspecific binding of amiodarone to albumin. Although we recognize that the use of PBS solution is not representative of real practice conditions, this choice was deliberate in order to explore specifically the potential interactions existing between amiodarone and the ECMO circuit.

In a complementary manner and beyond the in vitro results alone, the present study provides first evidence of an in vivo modification of amiodarone exposure in a male porcine ARDS model treated by VV ECMO and receiving amiodarone treatment after cardiac arrest. Indeed, males were chosen to avoid a possible hormonal effect on hemodynamics and distribution volumes, thus reducing experimental variability [21]. Although a recent systematic review reported only a small number of teams reporting experimental models of ARDS under VV ECMO published worldwide [22], the model developed in our study could be considered to evaluate the pharmacokinetics of drugs under VV ECMO during CPR. This is all the more true that the pharmacokinetic data of C_max_ and T_max_ found in our control group (C_max_ = 122.9 ± 51.8 mg/L; T_max_ = 63 ± 19 s) were of the same order of magnitude as those reported by Holloway et al. (C_max_ = 74.2 ± 33.1 mg/L; T_max_ = 94 ± 78 s) and Smith et al. (C_max_ = 64.1 ± 14.1 mg/L; T_max_ = 49 ± 21 s) in their models of cardiac arrest in pigs after administration of 300 mg intravenous amiodarone [11,12]. From this observation, we can postulate that the effect of ARDS on amiodarone concentrations in our experimental conditions and during the observation period appears to be negligible or very limited. Disposing of a validated experimental animal model before the first human clinical trial in such critical clinical conditions as cardiac arrest during ARDS could be a real advantage. Thus, the results obtained in the present work confirm and valid the need for further clinical studies to evaluate in humans the degree of bioavailability of amiodarone in patients receiving ECMO but also whether the 600 mg dose of amiodarone found in our study is able to compensate for the loss of amiodarone in the ECMO circuitry.

At the end of the in vivo study, we finally showed a decrease in C_max_ and AUC in the VV ECMO group compared with the control group. Since no significant difference was observed between the two groups with respect to T_max_, we can easily postulate that the observed decrease in AUC is mainly explained by the decrease in C_max_. It should be noted that accurate determination of T_max_ was supported by a very large number of blood samples (*n* = 8) over the first 5 min of the study. In addition to a probable adsorption phenomenon of amiodarone in the VV EMO, the reduction in plasma amiodarone concentrations could also be partially explained by a 10% increase in the volume of distribution due to priming of the ECMO circuit Indeed, the volume of 770 mL of 0.9% saline solution used in our study represented a non-negligible additional volume to the total blood volume of 4.8 L (75 mL/kg [23]) estimated in pigs. Moreover, physiological changes such as inflammation, known to increase the drugs distribution volume, have been previously described as a consequence of ECMO use [24]. Unfortunately, the inflammatory component was not investigated in our study but can be expected to be insignificant because of the very short duration of the experiment. Amiodarone clearance showed a 31% decrease driving a longer elimination half-life of amiodarone under VV ECMO which can be explained by the concept based on the mass balance equation, whereby a fluid is subject to the shared effects of cardiac output and VV ECMO pump flow [25]. A part of the IV amiodarone bolus is trapped in the VV ECMO circuitry and iteratively reinjected into the right atrium via the intrajugular canula, providing a slower elimination rate as well as lower amiodarone plasma concentrations. Finally, the decrease in amiodarone clearance in the ECMO group had a small impact on overall amiodarone exposure (AUC) compared with the associated effect of decreased C_max_ and increased volume of distribution.

Concerning the PK modeling used in this study, a two-compartment model was the best-fitted model for the amiodarone observed concentrations, which is consistent with current existing data. Indeed, although a three-compartment model is described as the best model describing the pharmacokinetics of amiodarone in long-term orally treated patients, a two-compartment model appears to fit better in patients treated with a single intravenous administration of amiodarone [26]. However, and in view of the predictive visual check of the final amiodarone model observed after the sixth minute, we admit that this PK model was not the best possible for predicting amiodarone concentration kinetics over the whole period of the experiment. Since the main change in exposure to amiodarone was found to be probably related to amiodarone adsorption leading to a decrease in C_max_, this limit of our study could be considered as a minor one. Another limitation in an attempt to design a more accurate PK model and also in the interpretation of the present study, was the small sample size of the population. This was especially true concerning the control group, where one out of the five pigs had to be as excluded from the statistical analysis due to its premature death during the experiment. However, the variability observed for C_max_ and T_max_ was within the range of those in the 28-pig studies by Holloway et al. and Smith et al. [11,12]. In addition, the variability of C_max_ and T_max_ in the VV ECMO group was surprisingly lower than in the control group and decreased with time. Although this effect of VV ECMO on interindividual variability of amiodarone concentrations needs to be further explored, this phenomenon strongly suggests the role of amiodarone adsorption in VV ECMO circuitry, which also depends on VV ECMO flow rate. Finally, we should note that part of the variability observed in our study may have resulted from the anatomic difference between the porcine and human sternum. Indeed, performing effective CPR in porcine model of cardiac arrest remains a challenge since the specific pectus carinatum of the porcine sternum, unlike the human sternum, did not perfectly fit the LUCAS automatic chest compression device.

In the face of the COVID-19 crisis, we focused on VV ECMO rather than on VA ECMO. Alterations in the pharmacokinetics of amiodarone should be greater with VV rather than VA ECMO, because there is only one transmembrane passage before systemic circulation under VA ECMO. Thus, our results should be considered only for VV ECMO. In addition, we did not evaluate the impact of membrane condition on amiodarone adsorption which depends on the duration of use of the membrane. Indeed, Dagan et al. reported that VV ECMO is associated with a significant reduction in concentrations of commonly used drugs, which partly depends on whether the membrane has been more or less recently implanted [16]. For example, these authors reported a 36% reduction in morphine concentration with a new membrane versus only 16% with a membrane used for 5 days. Moreover, our study evaluated the influence of VV ECMO on amiodarone pharmacokinetics after a single intravenous amiodarone injection. Therefore, nothing can be concluded about the effect of VV ECMO after repeated amiodarone injections. Although the present study showed a decrease in amiodarone exposure under VV ECMO conditions, we did not evaluate the effect of such a decrease on amiodarone efficacy. Finally, the data from our in vitro experiments as well as those obtained in pigs can be used to build a physiologically based pharmacokinetic (PBPK) model of amiodarone under VV ECMO. Indeed, PBPK models can address the limitations of traditional PK trials with compartmental modeling, by translating ECMO ex vivo results into dosing recommendations [27]. As recently illustrated for fluconazole, PBPK approaches provide a potent systematic way for making the most of already acquired knowledge immediately available in order to adapt the best suitable drug dosing to the need of patient on ECMO [28]. In order to model amiodarone exposure in patients on VV ECMO, a VV ECMO “organ” could be linked to the PBPK model and parameterized using data from the present study.

## 5. Conclusions

To the best of our knowledge, this study is the first reporting pharmacokinetics of amiodarone under VV ECMO in an animal model. At the end of experiment, we found a significant reduction in amiodarone exposure under VV ECMO conditions in a porcine ARDS model in cardiac arrest with ongoing CPR. Amiodarone doses greater than the usually recommended 300 mg, should be considered to reach an efficient amiodarone bioavailability under VV ECMO. A 600 mg dose of amiodarone should be tested in further clinical studies. The experimental protocol developed in the present study could be used to evaluate the influence of VV ECMO on other anti-arrhythmic drugs exposure, such as lidocaine. Finally, the results of the present study may be useful for the development of a PBPK model to assess amiodarone bioavailability in patients under VV ECMO.

## Figures and Tables

**Figure 1 pharmaceutics-14-00974-f001:**
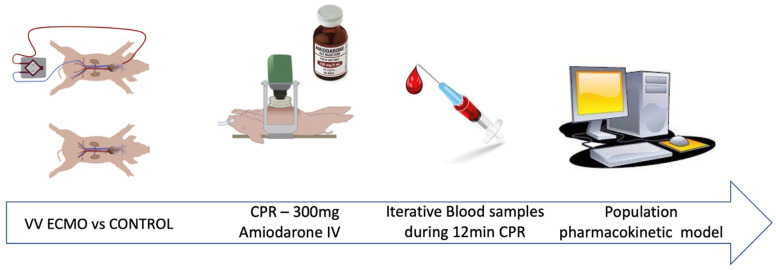
In vivo experimental protocol. ARDS: acute respiratory distress syndrome; VV ECMO: veno venous extracorporeal membrane oxygenation; CPR: cardiopulmonary resuscitation. IV: intravenous bolus.

**Figure 2 pharmaceutics-14-00974-f002:**
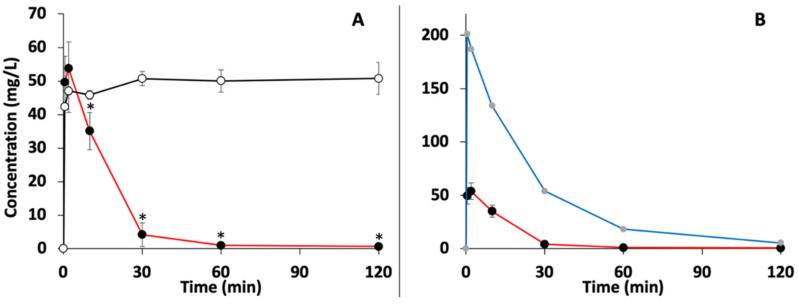
In vitro experiment. (**A**) Amiodarone concentration (mg/L) over time (min) measured in 100 mg amiodarone ECMO group (red line) (*n* = 3) and in 100 mg amiodarone control group (black line) (*n* = 3). (**B**) Amiodarone concentration (mg/L) over time (min) measured in 100 mg amiodarone ECMO group (red line) (*n* = 3) and in 300 mg amiodarone ECMO group (blue line) (*n* = 1). Data are presented as mean ± standard deviation. (*) statistically different in comparison to the control group (*p* < 0.05). Data are plotted for the first 2 h period. A very slight decrease in amiodarone concentrations was observed between 2 and 24 h (not shown).

**Figure 3 pharmaceutics-14-00974-f003:**
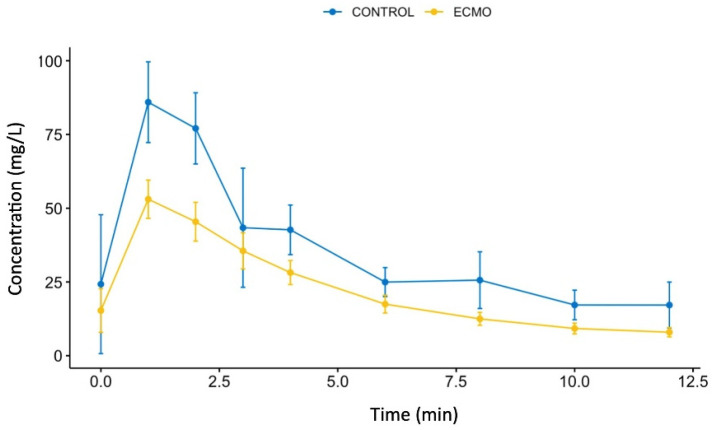
Evolution of amiodarone concentration after administration of a 300 mg amiodarone bolus along the in vivo experiment for ECMO group (yellow line) and control group (blue line) plotted as mean ± standard deviation.

**Figure 4 pharmaceutics-14-00974-f004:**
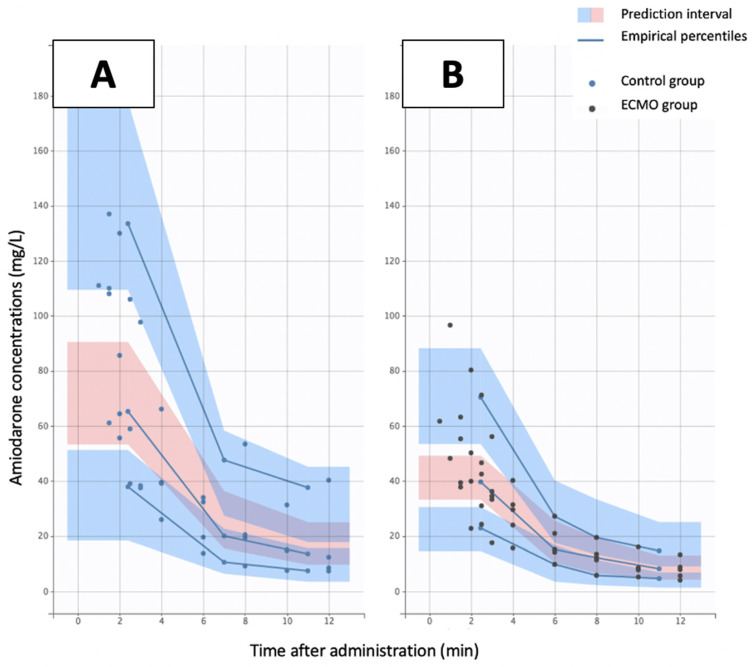
Predicted—Corrected Visual Predictive Check of the final amiodarone model. The observed data (blue spots for control group and black spots for VV ECMO group) were plotted with the median line, 10th and 90th percentiles of the predictions. The 90% confidence intervals of the median are represented by the pink shaded area. Data are presented for control (**A**) and ECMO (**B**) groups.

**Figure 5 pharmaceutics-14-00974-f005:**
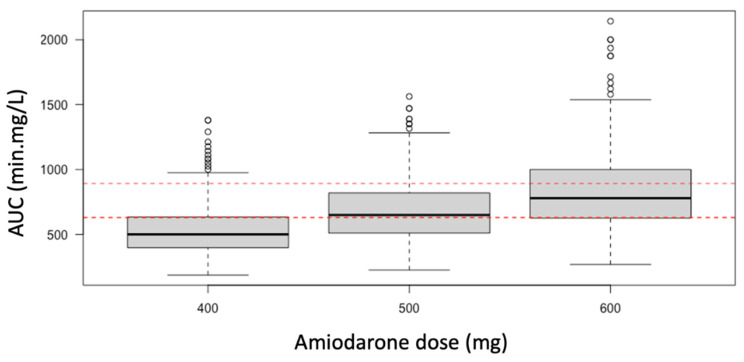
Simulated amiodarone exposure after administration of 400, 500 and 600 mg of amiodarone hydrochloride in the VV ECMO group. AUC: area under curve. Red dashed lines: amiodarone AUC Q1–Q3 in the control group. According to the Monte Carlo simulation, amiodarone 600 mg is required under VV ECMO for achieving the AUC control group.

**Table 1 pharmaceutics-14-00974-t001:** Body weight and hemodynamic parameters during cardiopulmonary resuscitation (CPR). Results are presented as median and interquartile range. No significant difference was observed between control and ECMO groups (*p* < 0.05).

Parameter	Control (*n* = 4)	ECMO (*n* = 5)	*p*-Value
Weight (kg)	65 (62–68)	67 (66–73)	0.805
Systemic Arterial Pressure (mmHg)			
Systolic	54 (47–59)	55 (54–65)	0.623
Mean	32 (31–34)	34 (32–35)	0.621
Diastolic	22 (20–25)	22 (16–24)	0.622
Carotid blood flow (mL/min)			
Mean	32 (22–54)	37 (19–60)	1

**Table 2 pharmaceutics-14-00974-t002:** Amiodarone pharmacokinetic parameters (T_max_, C_max_ and AUC_0__→__12min_) directly observed or calculated from amiodarone concentrations determined along the in vivo experiment. Data are presented as median and interquartile range. Statistical analysis was performed by Mann–Whitney test (*p* < 0.05).

Parameter	Control (*n* = 4)	ECMO (*n* = 5)	*p*-Value
T_max_ (s)	60 (56–67)	60 (45–75)	1
C_max_ (mg/L)	119.0 (97.1–144.7)	68.1 (55.3–80.3)	0.0635
AUC_0→12min_ (min·mg/L)	384 (354–438)	258 (240–288)	0.0159

**Table 3 pharmaceutics-14-00974-t003:** Estimated population parameters for amiodarone using a modeling approach. Between-Subject Variability (BSV) expressed as the coefficient of variation of the associated non-log-transformed parameter, clearance (Cl), central distribution volume (V_c_), peripheric compartment (V_p_), relative standard error (RSE) (standard error of the estimate divided by the estimate and multiplied by 100), impact of ECMO on clearance (β_ECMO/Cl_), impact of ECMO on central distribution volume (β_ECMO/Vc_), ω, coefficient of variation for between-subject variability; σ, parameters of error model.

Parameter	Model Mean	RSE (%)
Fixed effects		
Cl (L/h)	0.39	18.3
V_c_ (L)	1.09	28.5
β_ECMO/C__l_	0.69	34.7
β_ECMO/Vc_	1.10	29.9
Q (L/h)	0.33	10.4
V_p_ (L)	1.64	24.4
Between subject intervariability		
ωCl (%)	34.8	25.6
ωVc (%)	34.0	29.5
Residual variability		
σ (%)	18.3	9.2

**Table 4 pharmaceutics-14-00974-t004:** Individual pharmacokinetic parameters estimated by PK modeling (T_max_, C_max_ and AUC). Results are expressed as median and interquartile range. Significant difference was observed for maximum concentration (C_max_) between control and ECMO groups.

Parameter	Control (*n* = 4)	VV ECMO (*n* = 5)	*p*-Value
T_max_ (s)	90 (75–90)	90 (60–90)	1
C_max_ (mg/L)	123.5 (109.5–150.0)	61.7 (55.3–80.3)	0.015
AUC (min·mg/L)	698 (630–892)	400 (365–491)	0.063

## Data Availability

Available on request to the corresponding author.

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
