# Peer review of "Impaired Pharmacokinetics of Amiodarone under Veno-Venous Extracorporeal Membrane Oxygenation: From Bench to Bedside"

_pharmaceutics, 2022, doi:10.3390/pharmaceutics14050974_

Round 1

Reviewer 1 Report

This study described Pharmacokinetics of Amiodarone under Extracorporeal Membrane Oxygenation, which is very interesting topic. The methods are straightforward, and the results are clearly described. The result of in-vivo study was well supported by in-vitro experiment.

I have comments that would improve the paper.

  1. The AUC and Cmax were decreased in VV ECMO group and this phenomenon should be explained by increased clearance and volume of distribution of amiodarone in ECMO group. However, the parameters in Table 3 show that the CL is 0.69-fold decreased in the ECMO group. The authors should clarify it.
  2. I think there are some erroneous points in the final population pharmacokinetic model proposed in Table 3. The authors described that the final covariate model was, Cl = 0.39 × (0.69)^ECMO (line 365) and Vc = 1.09 × (1.1)^ECMO (line 367). If this is correct, the ECMO pigs will have low CL compared to the control groups, resulting in higher plasma concentration than control groups. I think the parameter estimates should be checked again.

Author Response

Dr Mickaël LESCROART

Intensive Care Unit

University Hospital of Nancy

Rue du Morvan

54511 Vandoeuvre-Les-Nancy Cedex, France

Tel: (+33) (0)6 13 67 37 09

E-mail: lescroart.mickael@gmail.com

                                                                                                     April, 25th 2022

Pharmaceutics

Dear reviewer,

We are thankful to the editors and the reviewers for their constructive comments and advices.

Please find enclosed the list of revisions and the revised version of our manuscript entitled “Impaired pharmacokinetics of amiodarone under veno-venous extracorporeal membrane oxygenation: from bench to bedside” for resubmission to Pharmaceutics. The manuscript was checked by a native Anglo-Saxon reviewer for the English language editing and all the Authors approve this new version of the original manuscript.

Yours sincerely,

Dr Mickael LESCROART

List of Revisions

Reviewer 1:

This study described Pharmacokinetics of Amiodarone under Extracorporeal Membrane Oxygenation, which is very interesting topic. The methods are straightforward, and the results are clearly described. The result of in-vivo study was well supported by in-vitro experiment. I have comments that would improve the paper.

  • Comment 1: The AUC and Cmax were decreased in VV ECMO group and this phenomenon should be explained by increased clearance and volume of distribution of amiodarone in ECMO group. However, the parameters in Table 3 show that the CL is 0.69-fold decreased in the ECMO group. The authors should clarify it.

Answer 1: We thank the reviewer for this comment. As stated in the first version of the manuscript, the decrease in AUC is mainly explained by the decrease in Cmax probably due to an adsorption phenomenon in the ECMO circuit, despite a decrease in the clearance of amiodarone in the ECMO group. Regarding Cmax, this parameter is mainly influenced by the change in dose and is not or only slightly influenced by the change in clearance. The observed decrease in amiodarone clearance could probably be explained by an iterative release of amiodarone from the ECMO pump. In conclusion, the decrease in amiodarone clearance in the ECMO group had a small impact on overall amiodarone exposure (AUC) compared with the associated effect of decreased Cmax and increased volume of distribution. To clarify this specific point, the discussion in the revised manuscript has been revised as follows:

Revised Manuscript – Line 519: “At the end of the in vivo study, we finally showed a decrease in Cmax and AUC in the VV ECMO group compared with the control group. Since no significant difference was observed between the two groups with respect to Tmax, we can easily postulate that the observed decrease in AUC is mainly explained by the decrease in Cmax. It should be noted that accurate determination of Tmax was supported by a very large number of blood samples (n=8) over the first 5 minutes of the study. In addition to a probable adsorption phenomenon of amiodarone in the VV EMO, the reduction in plasma amiodarone concentrations could also be partially explained by a 10 % increase in the volume of distribution due to priming of the ECMO circuit. Indeed, the volume of 770mL of 0.9% saline solution used in our study represented a non-negligible additional volume to the total blood volume of 4.8L (75mL/kg [23]) estimated in pigs. Moreover, physiological changes such as inflammation, known to increase the drugs distribution volume, have been previously described as a consequence of ECMO use [24]. Unfortunately, the inflammatory component was not investigated in our study but can be expected to be insignificant because of the very short duration of the experiment. Amiodarone clearance showed a 31% decrease driving a longer elimination half-life of amiodarone under VV ECMO which can be explained by the concept based on the mass balance equation, whereby a fluid is subject to the shared effects of cardiac output and VV ECMO pump flow [25]. A part of the IV amiodarone bolus is trapped in the VV ECMO circuitry and iteratively reinjected into the right atrium via the intrajugular canula, providing a slower elimination rate as well as lower amiodarone plasma concentrations. Finally, the decrease in amiodarone clearance in the ECMO group had a small impact on overall amiodarone exposure (AUC) compared with the associated effect of decreased Cmax and increased volume of distribution.”

  • Comment 2: I think there are some erroneous points in the final population pharmacokinetic model proposed in Table 3. The authors described that the final covariate model was, Cl = 0.39 × (0.69)^ECMO (line 365) and Vc = 1.09 × (1.1)^ECMO (line 367). If this is correct, the ECMO pigs will have low CL compared to the control groups, resulting in higher plasma concentration than control groups. I think the parameter estimates should be checked again.

Answer 2: As explained in the previous answer (Answer 1), the clearance and volume of distribution of amiodarone were lower (31% decrease) and higher (10% increase) in the ECMO group, respectively. The explanation for these observations is discussed in the previous answer (Answer 1).

- For the clearance of the group without ECMO, the expression of the clearance Cl = 0.39 × (0.69)ECMO became:

Cl = 0.39 × (0.69 )0 = 0.39 x 1 = 0.39 L/h in the non-ECMO group.

- For the clearance of the group with ECMO, the expression of the clearance Cl = 0.39 × (0.69)ECMO became:

Cl = 0.39 × (0.69)1 = 0.39 x 0.69 = 0.27 L/h in the ECMO group.

Clearance in the ECMO group is therefore lower than clearance in the non-ECMO group by 31% : (0.39-0.27)/0.39 = 31%.

Likewise for volume:

- For the volume of the group without ECMO, the expression is V = 1.09 × (1.1)ECMO became:           V = 1.09 × (1.1)ECMO = 1.09 x 1.10 = 1.09 x 1 = 1.09 L in the non-ECMO group.

- For the volume of the group with ECMO, the expression is V = 1.09 × (1.1) ECMO became: V = 1.09 × (1.1)ECMO = 1.09 x 1.11 = 1.09 x 1.1 = 1.2 L in the ECMO group.

The volume in the ECMO group is therefore higher than the volume in the non-ECMO group by 10% (1.09-1.2)/1.09= 10%.

A specific mention was added in the text as follow line 396 :

“Clearance was 0.39 L/h and 0.27 L/h in the non-ECMO and ECMO group respectively, resulting in a 31% lower clearance in the ECMO group.

Similarly, central distribution volume was 1.09 L and 1.2 L, in the non-ECMO and ECMO group respectively, resulting in a 10% increased distribution volume in the ECMO group.”

Reviewer 2 Report

The research manuscript describes work based on impaired pharmacokinetics of Amiodarone under Veno-Venous Extracorporeal Membrane Oxygenation. The manuscript is well structured and the conclusion is drawn from different experimental data. However, there are the following minor flaws that may be addressed by the authors.

Decision: Minor revision

  1. Introduction section is very brief. Please give details about the rationale, objective, hypothesis, and background of the work with relevant references.
  2. Figure 2 B shows that the group with 300mg amiodarone ECMO does not have an error bar. Please justify statistical relevance.
  3. For animal study authors have used male pigs. Please describe why female animals were not preferred. Also, have the authors run a power analysis to find out the number of animals per group? How the number of animals per group was decided so that the data can give statistical significance?

Author Response

Dr Mickaël LESCROART

Intensive Care Unit

University Hospital of Nancy

Rue du Morvan

54511 Vandoeuvre-Les-Nancy Cedex, France

Tel: (+33) (0)6 13 67 37 09

E-mail: lescroart.mickael@gmail.com

                                                                                                          April, 25th 2022

Pharmaceutics

Dear reviewer,

We are thankful to the editors and the reviewers for their constructive comments and advices.

Please find enclosed the list of revisions and the revised version of our manuscript entitled “Impaired pharmacokinetics of amiodarone under veno-venous extracorporeal membrane oxygenation: from bench to bedside” for resubmission to Pharmaceutics. The manuscript was checked by a native Anglo-Saxon reviewer for the English language editing and all the Authors approve this new version of the original manuscript.

Yours sincerely,

Dr Mickael LESCROART

List of Revisions

____________________________

Reviewer 2:

  • Comment 1: Introduction section is very brief. Please give details about the rationale, objective, hypothesis, and background of the work with relevant references.

Answer 1: We thank the reviewer for this comment. The introduction of the revised manuscript has been modified as follows to address the background data and specific assumptions underlying the study:

Revised Manuscript – Line 48: “The use of veno-venous extracorporeal membrane oxygenation (VV ECMO) has increased since the 2009 H1N1 pandemic, and might improve survival up to 70 % in COVID-19 acute respiratory distress syndrome (ARDS) [1]. Nevertheless, VV ECMO was made responsible for impairing drug pharmacokinetics (PK), due to a possible modification of distribution volumes as well as to potential physicochemical interactions between drugs and ECMO circuitry, especially in case of highly lipophilic molecules [2]. Since the 1970s, membrane oxygenators (MO) have been progressively modified to make them smaller and more efficient, and their compounds now consist of molecules such as polymethylpentane (PMP) instead of silicone. PMP is a microporous polymer with a thin non-porous matrix on the blood side that requires diffusion and pressure gradients for molecules to pass through, which would minimize the amount of plasma leakage through the membrane. Harthan et al. reported that, despite the use of newer components in ECMO circuits, a large amount of medication is adsorbed into the circuit [3]. Absorption is driven by the electrostatic and hydrophobic interactions. Electrostatic interactions dominate when surface coatings are applied to the circuit, while the hydrophobic interactions tend to dominate when lipophilic drugs adhere to the tubes and membrane oxygenator without coating [4]. Wildschut et al. demonstrated that drug absorption is positively correlated with the degree of lipophilicity and that octanol/water partition coefficient (logP) values could predict increased drug loss [5]. The effect of ECMO on drugs PK has been studied for sedative and anti-infective medications and Raffaeli et al. reported in their in vitro study a substantial drug loss in the extracorporeal membrane oxygenation circuits within 24 hours of use, for the following molecules: paracetamol 49%, morphine 51%, midazolam 40%, fentanyl 84%, sufentanil 83% [6]. Since amiodarone is one of the most lipophilic drugs (LogP = 7.58), the issue of a possible influence of ECMO circuits on its bioavailability may rise.

To the best of our knowledge, only pediatric sparse case reports have shown therapeutic failure of its delivery at conventional posology during shockable cardiac arrest, whereas amiodarone PK under VV ECMO has never been studied in vivo so far [7,8]. It is a matter of the utmost importance because amiodarone is one of the very few molecules that can be safely used in intensive care units for treating both cardiac arrhythmias and shockable cardiac arrest [9]. The incidence of paroxysmal atrial fibrillation (AF) is high in critically ill patients [10]. This atrial arrhythmia has been associated with worsen outcome in patients with ARDS [11]. The loss of atrial function induces a 20% drop in cardiac output, which impairs oxygen delivery [12]. Recently, Li et al. reported that new-onset atrial arrhythmias are a frequent complication during VV ECMO and are independently associated with odds ratio for in-hospital mortality as high as 2.21 CI95 [1.08 – 4.55], with an interesting early temporal association of atrial arrhythmias with ECMO initiation (median time to onset of 1.7 days after ECMO deployment) [13,14]. Otherwise, the worst conditions clinicians would have to manage are refractory ventricular arrhythmias and cardiac arrest. According to the 2018 American Heart Association (AHA) guidelines for cardiopulmonary resuscitation (CPR), intravenous (IV) amiodarone 300 mg bolus should be administered during shockable cardiac arrest refractory to 3 consecutive electrical cardioversions, occurring apart from VV ECMO conditions [15]. In addition, Mc Daniel et al. showed in their ex vivo model, a rapid and heavy extraction of amiodarone by ECMO circuitry during the first 30 minutes of the procedure [16]. We postulated that CPR under VV ECMO would represent the worst impairment of amiodarone PK although this situation requires early high concentrations at the aortic root to ensure efficient receptor-ligand interactions and to improve the pharmacodynamics (PD) of amiodarone. In line with these observations, the present study aimed to assess the role of VV ECMO on amiodarone PK using both an in vitro model and an in vivo ARDS porcine CPR model.”

  • Comment 2: Figure 2 B shows that the group with 300mg amiodarone ECMO does not have an error bar. Please justify statistical relevance.

Answer 2: We thank the reviewer for this remark. The experiment was performed once for the 300mg dose of amiodarone. No standard deviation could be estimated from a single experiment, although the results were consistent with the 100mg experiment. This was specifically mentioned in page 3 line 242: “The experiment was performed thrice for amiodarone 100 mg and control groups, and once for the amiodarone 300 mg group.” And in the title of Figure 2 B : “measured in 100mg amiodarone ECMO group (red line) (n=3) and in 300mg amiodarone ECMO group (blue line) (n=1)”.

  • Comment 3: Methods : Please describe why female animals were not preferred.

Answer 3: All the pigs were male to avoid a possible hormonal effect on hemodynamics and volumes of distribution. Exposure to estrogen and progesterone has important effects on both body fluid regulation and cardiovascular function, and both of these reproductive hormones impact blood pressure responses to sodium loads. A specific statement was added in the revised manuscript as follow:

Revised Manuscript – Line 500: “Indeed, males were chosen to avoid a possible hormonal effect on hemodynamics and distribution volumes, thus reducing experimental variability [21].”

  • Comment 4: Methods: have the authors run a power analysis to find out the number of animals per group ?

Answer 4: We thank the reviewer for this remark. Due to the experimental and pilot nature of the study, it was not possible to perform a sample size calculation. This statement was added in the revised manuscript as follow:

Revised Manuscript – Line 278: “Due to the experimental and pilot nature of the study, it was not possible to perform a sample size calculation.”
